# Derivation of the Immortalized Cell Line UM51-PrePodo-hTERT and Its Responsiveness to Angiotensin II and Activation of the RAAS Pathway

**DOI:** 10.3390/cells12030342

**Published:** 2023-01-17

**Authors:** Lars Erichsen, Lea Doris Friedel Kloss, Chantelle Thimm, Martina Bohndorf, Kira Schichel, Wasco Wruck, James Adjaye

**Affiliations:** 1Institute for Stem Cell Research and Regenerative Medicine, Medical Faculty, Heinrich-Heine University Duesseldorf, 40225 Duesseldorf, Germany; 2EGA Institute for Women’s Health, University College London, 86-96 Chenies Mews, London WC1E 6HX, UK

**Keywords:** podocytes, renin–angiotensin-system, human telomerase

## Abstract

Recent demographic studies predict there will be a considerable increase in the number of elderly people within the next few decades. Aging has been recognized as one of the main risk factors for the world’s most prevalent diseases such as neurodegenerative disorders, cancer, cardiovascular disease, and metabolic diseases. During the process of aging, a gradual loss of tissue volume and organ function is observed, which is partially caused by replicative senescence. The capacity of cellular proliferation and replicative senescence is tightly regulated by their telomere length. When telomere length is critically shortened with progressive cell division, cells become proliferatively arrested, and DNA damage response and cellular senescence are triggered, whereupon the “Hayflick limit” is attained at this stage. Podocytes are a cell type found in the kidney glomerulus where they have major roles in blood filtration. Mature podocytes are terminal differentiated cells that are unable to undergo cell division in vivo. For this reason, the establishment of primary podocyte cell cultures has been very challenging. In our present study, we present the successful immortalization of a human podocyte progenitor cell line, of which the primary cells were isolated directly from the urine of a 51-year-old male. The immortalized cell line was cultured over the course of one year (~100 passages) with high proliferation capacity, endowed with contact inhibition and P53 expression. Furthermore, by immunofluorescence-based expression and quantitative real-time PCR for the podocyte markers CD2AP, LMX1B, NPHS1, SYNPO and WT1, we confirmed the differentiation capacity of the immortalized cells. Finally, we evaluated and confirmed the responsiveness of the immortalized cells on the main mediator angiotensin II (ANGII) of the renin–angiotensin system (RAAS). In conclusion, we have shown that it is possible to bypass cellular replicative senescence (Hayflick limit) by TERT-driven immortalization of human urine-derived pre-podocyte cells from a 51-year-old African male.

## 1. Introduction

Podocytes are a distinct cell type within the kidney glomerulus. Their major task is the filtration of blood to generate urine, and thereby retaining plasma proteins [1]. Therefore, they are part of the glomerular filtration barrier, together with glomerular epithelial cells (GECs) and the glomerular basement membrane (GBM) [2]. To execute this complex task, podocytes need to be terminally differentiated and develop an elaborate and highly specialized actin cytoskeleton. A typical podocyte morphology is composed of three major segments, the cell body, major processes and foot processes (FPs) [3]. Major processes are connected to the cell body of podocytes and can split into foot processes. Neighboring foot processes from different podocytes form an inter-digitating pattern, which is bridged by slit diaphragms (SDs) [4].

During development of the human kidney, the precursor cells of podocytes arise from SIX2-positive renal progenitor cells. Subsequent developmental processes are then needed to acquire the terminally differentiated state and maturity [4,5]. Glomerular development proceeds in four stages: the renal vesicle stage, the S-shaped body stage, the capillary loop stage, and the maturing-glomeruli stage [6,7]. During the S-shaped body stage, podocyte precursors are simply undifferentiated epithelial cells, with apically located tight junctions [8] and prominent mitotic activity [9]. During this specific time point, expression of podocalyxin (PODXL) [10] and the tight junction protein ZO-1 [11] are initiated, while expression of the podocyte transcription factor WT1 is the highest expression level [12]. When these immature podocytes enter the capillary loop stage, their mitotic activity is abolished and they begin to establish their complex actin cytoskeleton, including the appearance of FPs and the reorganization of cell–cell junctions into glomerular slit diaphragms (SD) [8]. The filtration slits are formed by the spatial formation of the FPs. These are spanned by the SD, which is similar to an adherens junctions structure [13]. For the formation of SD, the protein nephrin (NPHS1) is required, which is additionally associated with the actin cytoskeleton and, thus, contributes to the actin dynamics of podocytes and the formation of FPs [14].

The phenotypical changes occurring between the S-shaped body stage and the capillary loop stage are accompanied by synaptopodin (SYNPO) [15] and vimentin [16] expression. Therefore, both have been recognized as key markers of differentiated post-mitotic podocytes [15,17,18], which cannot be detected in undifferentiated or dedifferentiated cells [19,20].

Numerous kidney diseases, such as chronic kidney disease, membranous nephropathy, congenital nephrotic syndrome and Alport syndrome are associated with proteinuria and or hematuria, which is mainly caused by defects in the GBM or alterations in the structure and or function of the podocytes. Modelling of these pathological conditions has been very challenging. One obvious drawback of studying human podocytopathies is the limited proliferation capacity of these cells. This was further aggravated by the fact that the derivation of kidney-originating cells is very challenging. This is particularly true for podocytes since their complex architecture is not well preserved from kidney biopsy tissue [17]. For these reasons, disease modelling of human kidney alterations is often studied in animal models which only replicate human conditions to a limited extent. Therefore, other models have been implemented to study human kidney disease, such as the temperature-sensitive SV40 conditionally immortalized podocyte cell line [17] (UdRPCs, Rahman et al.; and many iPSC-based differentiation protocols for cells of kidney origin, Erichsen et al. [21,22,23,24,25]).

Telomeres are GT-rich repeats (TTAGGG)_n_ at chromosome ends in human and other vertebrates [26,27]. These structures are established by the enzyme telomerase. Telomerase is an RNA-dependent DNA polymerase [28] consisting of two essential components. The first is the functional RNA component (*hTR* or *hTERC*), which serves as template for telomere synthesis, and the second is the catalytic domain with reverse transcriptase activity (hTERT) [29,30,31,32]. While pluripotent stem cells possess high telomerase activity, the level drops during differentiation into somatic cells. hTERT remains active in some tissues (e.g., germ cells) and a selection of cell populations [33]. The activity is regulated to meet the demand of proliferation to facilitate specific cellular functions, while simultaneously serving as a barrier against carcinogenesis by inducing senescence [34,35]. Numerous mammalian and human cell types have been immortalized by using either hTERT transfection [36,37,38,39] or by the introduction of viral oncogenes, such as SV40 Large-T antigen, adenovirus E1A and E1B, papilloma virus E6 and E7, CELO virus orf22 and GAM-1. In contrast with hTERT transfections, the main drawback of these viral oncogenes is that the generated cell lines often lose cell cycle and apoptosis control due to the inhibition of the pRB and p53 pathways, respectively, which ultimately results in malignant cell transformation [37,40].

Here, we describe the successful immortalization of a human podocyte progenitor cell line, obtained directly from urine. The cells showed a stable morphology and proliferation capacity in vitro for more than one year in culture. They expressed typical podocyte markers, such as SYNPO, NPHS1 and WT1, and upon our recently reported differentiation protocol [41], their proliferation capacity was retained, and they expressed several podocyte markers such as SYNPO. Due to their high proliferation capacity and their ease of handling, our immortalized UM51-hTERT-Pre-Podo cell line offers a unique tool for studying nephrogenesis and kidney-associated diseases.

## 2. Material and Methods

### 2.1. Cell Culture Conditions

The cell line UM51-PrePodo was derived from the urine of a 51-year-old male of African origin. The cells were cultured, and differentiation was induced as described in [41]. Immortalization of the cells was achieved by cationic polymer transfection of the pCDNA-3xHA-hTERT plasmid with Xfect (Takara BIO INC, Kusatsu, Shiga Prefecture, Japan). The plasmid pCDNA-3xHA-hTERT was obtained from (Addgene plasmid # 51637; http://n2t.net/addgene:51637; RRID: Addgene_51637; accessed 12 January 2023) [42]. In brief, 2 µg of the plasmid was incubated at RT for 10 min with 100 µL of transfection buffer and 1 µL of transfection reagent. After incubation, the mix was added to at least 50% confluent 6 well of UM51-PrePodo, growing as a monolayer. In another transfection, 2 µg of the commercial pmaxGFP Vector (Lonza, Basel, Switzerland) was incubated at RT for 10 min with 100 µL of transfection buffer and 1 µL of transfection reagent. After incubation, the mix was added to at least 50% confluent 6 well of UM51-PrePodo-hTERT, growing as monolayer.

### 2.2. Immunofluorescence-Based Detection of Protein Expression

Cells were fixed with 4% paraformaldehyde (PFA) (Polysciences, Warrington, FL, USA). Unspecific binding sites were blocked by incubation with blocking buffer, containing 10% normal goat or donkey serum, 1% BSA, 0.5% Triton, and 0.05% Tween, for 2 h at room temperature. Incubation of the primary antibody was performed at 4 °C overnight in staining buffer (blocking buffer diluted 1:1 with PBS). After at least 16 h of incubation, the cells were washed three times with PBS/0.05% Tween and incubated with a 1:500 dilution of secondary antibodies. After three additional washing steps with PBS/0.05% Tween, the cell and nuclei were stained with Hoechst 1:5000 (Thermo Fisher Scientific, Waltham, MA, USA). Images were captured using a fluorescence microscope (LSM700; Zeiss, Oberkochen, Germany) with Zenblue software (Zeiss). Individual channel images were processed and merged with Fiji. Detailed information of the used antibodies is given in Appendix A.

### 2.3. Western Blotting

Protein isolation was performed by lysis of the cells in RIPA buffer (Sigma-Aldrich, St. Louis, MO, USA), supplemented with complete protease and phosphatase inhibitors (Roche, Basel, Switzerland). Proteins were resolved on a 7.5% Bis-Tris gel and blotted onto a 0.45 µm nitrocellulose membrane (GE Healthcare Life Sciences, Chalfont St. Giles, UK). After blocking the membranes with 5% skimmed milk in Tris-buffered Saline Tween (TBS-T), they were incubated overnight with the respective primary antibodies (Appendix A). Membranes were washed 3× for 10 min with TBS-T. Secondary antibody incubation was performed for 1 h at RT, and membranes were subsequently washed 3× for 10 min with TBS-T. Amersham ECL Prime Western Blotting Detection Reagent was used for the chemiluminescent detection (GE Healthcare Life Sciences) and captured with the imaging device Fusion FX.

### 2.4. Southern Blotting

Genomic DNA isolation was performed with the DNeasy^®^ Blood & Tissue Kit (Qiagen, Hilden, Germany) according to the manufacturer’s instructions. An amount of 10 µg of the isolated DNA was digested by incubation with EcoRI and HindIII overnight and separated on a 1% agarose gel for 2 h. Prior to blotting of the DNA onto a positively charged nylon membrane (GE Healthcare Life Sciences), the in-gel DNA was denaturated by incubation in 0.25 M HCL solution for 5 min, followed by 2×/15 min in denaturation solution and 2×/15 min in neutralization solution. DNA transfer was carried out overnight at room temperature with 10× saline-sodium citrate (SSC) buffer. Subsequently the membrane was fixed at 120 °C for 20–35 min and washed twice with SSC. Telomere fragments were detected by a prehybridization step with DIG Easy Hyb Granules for 1 h and by the actual hybridization of the membrane with a solution containing Telomere Probe dissolved 1:5000 in DIG Easy Hyb granules (Sigma Aldrich) at 42 °C overnight. Two stringent washing steps were performed after hybridization, blocking was performed in blocking solution followed by Anti-DIG antibody solution. For detection, the membranes were washed in detection buffer for 5 min, and 1 mL substrate solution was added. Pictures were taken with the imaging device Fusion FX after an incubation period of at least 5 min.

### 2.5. Relative Quantification of Podocyte-Associated Gene Expression by Real-Time PCR

Real-time PCR of podocyte-associated gene expression was performed as follows.

An amount of 1 µg of RNA was used as input for reverse transcription in a volume of 10 µL. Real-time measurements were carried out on the Step One Plus Real-Time PCR Systems using 1:10 diluted template cDNA in a MicroAmp Fast Optical 384 Well Reaction Plate and Power Sybr Green PCR Master Mix (Applied Biosystems, Foster City, CA, USA). The amplification conditions were denatured at 95 °C for 13 min followed by 37 cycles of 95 °C for 50 s, 60 °C for 45 s and 72 °C for 30 s. Primer sequences are listed in Appendix A.

### 2.6. Resazurin Assay

Proliferation of UM51-PrePodo cells grown in proliferation medium and advanced RPMI was measured with a resazurin assay (Sigma-Aldrich) according to the manufacturer’s instructions.

In brief, these were: Per condition (PM/adv. RPMI) 50,00 cells were seeded in 500 µL of the respective medium in triplicate. An amount of 10× Resazurin solution was mixed with the two media in a 1:10 dilution, and cells were incubated at 37 °C and 5% CO_2_ for 2 h. An amount of 200 µL medium for each condition was transferred into a 96-well plate and absorbance at 570 and 600 nm was measured with Plate Reader AF2200 (Eppendorf, Hamburg, Germany).

### 2.7. Statistics

Data are presented as arithmetic means and standard error. In total, three independent experiments were performed and used for the calculation of mean values. Statistical significance was calculated using the two sample Student’s *t*-test with a significance threshold of *p* = 0.05.

## 3. Results

### 3.1. Successful Immortalization of UM51-PrePodo Primary Cells

UM51-PrePodo cells were isolated directly from human urine as described in [43]. The cells were transfected with the plasmid pCDNA-3xHA-hTERT [42]. Immortalization was achieved after four months, and the cells referred to as UM51-PrePodo-hTERT. This cell line exhibited a much faster proliferation capacity compared with the original primary cells UM51-PrePodo, and kept proliferating for more than six months, while showing a similar morphology (Figure 1A). Both cell lines grew as monolayers, showed contact inhibition and a cobblestone-like morphology typical for epithelial cells. We further analyzed P53 expression and phosphorylation by immunofluorescence-based protein detection (Figure 1B). While most cells expressed P53, only a small percentage were positively stained for phosphorylated P53. Since the transfected hTERT protein was N-terminally tagged with 3× HA, successful transfection of the cells was evaluated by Western blotting and immunocytochemistry with a primary antibody recognizing the HA-tag (Figure 1C,E). Generally, the transformed cells tended to appear a little smaller and more elongated than UM51-PrePodo. Since ectopic expression of hTERT has been associated with chromosomal alterations and genomic instabilities [44,45,46,47], we analyzed the karyotype of UM51-PrePodo-hTERT (Figure 1D). In total, 20 metaphases were analyzed and a hypertriploid karyotype with several chromosomal alterations was observed in all of them. Since alterations of this magnitude might have an impact on transcriptional activity, we could not exclude the probability of cancerous transformation. The immunofluorescence-based analyses detected expression of the HA-tag in UM51-PrePodo-hTERT. Furthermore, Western blot analysis revealed a band at approximately 80 kDa in the protein lysate of UM51-PrePodo-hTERT, which was absent in the non-transfected cell line UM51-PrePodo. ß-Actin was used as loading control and detected at size of 45 kDa (full size Western blot images are given in Appendix A). In addition, hTERT gene expression was determined by quantitative real-time PCR, revealing a 132-fold increase in mRNA expression (Figure 1F). The telomere length of UM51-PrePodo-hTERT cells in comparison with the primary UM51-PrePodo cells was evaluated by Southern blotting. Terminal restriction fragments (TRF) containing telomeres and a sub-telomeric region were detected in the control DNA originating from an immortal cell line as supplied by Sigma Aldrich, the induced pluripotent stem cell line A4, as well as the UM51-PrePodo and UM51-PrePodo-hTERT (Figure 1G). TRFs of the A4 cell line were found to be the longest at approximately 13.9 kbp, followed by the control with a fragment size of 7.4 kbp, UM51-PrePodo of 6.3 kbp, and UM51-PrePodo-hTERT had the shortest TRFs with a length of 1.2 kb.

### 3.2. Characterization UM51-PrePodo-hTERT

To determine the effect of hTERT-driven immortalization of the primary UM51-PrePodo cells on their differentiation capacity, we performed immunofluorescence-based detection of WT1 and NPHS1 protein and real-time PCR based detection of mRNA expression for the genes *WT1*, *NPHS1*, *CD2AP*, *CD24* and *CD106* (Figure 2A–E). As indicated by the immunofluorescence staining, the primary UM51-PrePodo cells, as well as the immortalized UM51-PrePodo-hTERT cell line, both expressed key podocyte proteins WT1 and NPHS1 (Figure 2A–D). The comparison between the primary and the immortalized cell line by real-time PCR based detection revealed a significant increase in mRNA expression for *WT1*, *NPHS1* and *CD106*, by 0.77 (*p* = 0.001), 2.83 (*p* = 0.03) and 1.66-fold (*p* = 0.05), respectively, and a significant downregulation of *CD2AP* and *CD24*, by 0.57 (*p* = 0.05) and 0.52 (*p* = 0.05) in the immortalized cells. To test transfection efficacy of the UM51-PrePodo-hTERT cell line, we cationic polymer transfected freshly plated cells at approximately 60–70% confluency with the pmaxGFP Vector, and obtained 70% GFP positive cells after 24 h (Appendix A).

### 3.3. Culturing UM51-PrePodo-hTERT in adv. RPMI Initiates Podocyte Differentiation

We recently reported a detailed protocol for the differentiation of SIX2-positive urine derived renal progenitor cells (UdRPCs) into mature podocytes [41]. We applied this protocol to differentiate the UM51-PrePodo-hTERT cell line into podocytes. To determine the proliferation rate of UM51-PrePodo-hTERT cultured under the differentiation conditions, we applied a resazurin assay for 72 h (Figure 3A). The fold-change of normalized absorbance was found to increase in UM51-PrePodo-hTERT cultured in PM medium by 0.8-fold after 48 h and by additional 0.74-fold after 72 h. In contrast, UM51-PrePodo-hTERT cultured in adv. RPMI medium showed only minor changes in absorbance with 0.03-fold increase after 48 h and 0.13-fold increase after 72 h. The difference in proliferation rate was also observed using light microscopy (Appendix A). While UM51-PrePodo-hTERT cells cultured in proliferation medium were at 100% confluency after 72 h, cells cultured in advanced RPMI were approximately 60% confluent. Relative protein expression normalized to ß-ACTIN for the HA-tag, for the key podocyte marker SYNPO as well as P53, were detected by Western blotting (Figure 3B). All proteins were detected at the expected sizes of 80 kDa for the HA-tag, 50 kDA for LMX1B, 100 kDA for NPHS1, 110 kDa for SYNPO, 53 kDa for p53, and 45 kDa for ß-ACTIN. While the most significant difference in protein expression was observed for SYNPO with a 4.8-fold increase (*p* = 0.04), HA-tag expression decreased with a fold change of 0.6 (*p* = 0.05). Interestingly, the protein level of P53 was unaffected by the culturing medium (full size Western blot images are given in Appendix A). Expression of the differentiated podocyte marker SYNPO and the proliferation marker KI-67 were assessed by immunofluorescence-based protein detection (Figure 3C–F). KI-67-protein was detected in the nuclei of UM51-PrePodo-hTERT cells cultured in PM (Figure 3C) as well as adv. RPMI medium (Figure 3D). As expected, a lower percentage of cells grown in adv. RPMI expressed the proliferation marker than cells cultured in PM—3.8% compared with 13.9%, respectively (Figure 3G). While UM51-PrePodo-hTERT cells grown in PM lacked SYNPO protein expression (Figure 3E), a clear signal could be detected in cells grown in adv. RPMI (Figure 3F). Additionally, we measured the expression of *CD24, CD106, SYNPO, P53* and *KI67* by quantitative real-time PCR (Figure 3H). In accordance with the Western blot and immunofluorescence-based protein detection, the mRNA level of *SYNPO* was found to be significantly increased (3.2-fold; *p* = 0.01), while *P53* and *KI67* were found to be downregulated (0.41, *p* = 0.01 and 0.82-fold, *p* = 0.01, respectively). Furthermore, mRNA expression of the surface marker CD106 was found to be significantly downregulated (0.74-fold, *p* = 0.04) and CD24 expression was found to be unaltered.

### 3.4. Comparative Transcriptome Analysis of Urine Derived Renal Progenitor Cells UM51 with the Immortalized UM51-PrePodo-hTERT Cell Line

After the successful immortalization of UM51-PrePodo into UM51-PrePodo-hTERT and differentiation with adv. RPMI, we performed a comparative transcriptome analysis. Hierarchical clustering analysis comparing the transcriptomes of UM51-PrePodo and UM51 podocytes with their respective immortalized counterpart revealed a distinct expression pattern of wild type and immortalized cells (Figure 4A). By comparing the expressed genes (det-p  <  0.05), we found that 582 were exclusively expressed in the UM51-PrePodo and 703 in UM51-PrePodo-hTERT grown in PM (Figure 4B). In total, 13,914 genes were found to be expressed in common between UM51-PrePodo and UM51-PrePodo-hTERT grown in PM (Figure 4B). After differentiation of UM51-PrePodo and UM51-PrePodo-hTERT by culturing the cells in adv. RPMI supplemented with 30 µM retinoic acid, transcriptome data revealed 880 genes to be exclusively expressed in UM51 podocytes and 460 in UM51-PrePodo-hTERT grown in adv. RPMI (Figure 4C). In total, 13,922 genes were found to be expressed in common between UM51-PrePodo-hTERT grown in adv. RPMI and UM51-PrePodo-hTERT grown in PM (Figure 4D). In total, 13,699 genes were found to be expressed in common between UM51-Podocytes and UM51-PrePodo-hTERT grown in adv. RPMI (Figure 4C). By comparing the expressed genes (det-p  <  0.05), 237 were exclusively expressed in the UM51-PrePodo-hTERT grown in adv. RPMI and 695 in UM51-PrePodo-hTERT grown in PM (Figure 4D). Wild-type cells and their immortalized counterparts share many of the most over-represented GO BP-terms, such as regulation of ion transport, regulation of organic acid transport and developmental processes. The most over-represented GO BP-terms exclusive to UM51-PrePodo were associated with planar cell polarity involved in axis elongation and skin development. In comparison, the most over-represented GO BP-terms exclusively expressed in the immortalized UM51-PrePodo-hTERT cells grown in PM were associated with cell-cycle checkpoints, nuclear division, regulation of mitotic cycle and DNA replication (Figure 4E). The most over-represented GO BP-terms exclusive to UM51 podocytes were associated with regulation of ion transport, heterophilic cell–cell adhesion via plasma membrane cell adhesion molecules, and regulation of system processes. In comparison, the most over-represented GO BP-terms exclusively expressed in the immortalized UM51-PrePodo-hTERT cells grown in adv. RPMI were associated with chromosome segregation, cell-cycle checkpoints, DNA replication, cell cycle and regulation of DNA replication (Figure 4F). The full gene list can be found in Appendix A. UM51-PrePodo-hTERT cells grown in both media shared many of the most over-represented GO BP-terms, such as cell junction organization, urogenital system development, potassium ion transport and GPCR ligand binding. The most over-represented GO BP-terms exclusive to UM51-PrePodo-hTERT grown in adv. RPMI were associated with specification of animal organ identity and chemotaxis. In comparison, the most over-represented GO BP-terms exclusively expressed in the immortalized UM51-PrePodo-hTERT cells grown in PM were associated with embryonic pattern specification, embryonic organ development and positive regulation of cell–cell adhesion (Figure 4G). The full gene list can be found in Appendix A. Additionally, the transcriptomes of UM51-PrePodo and the immortalized UM51-PrePodo-hTERT grown in PM and adv. RPMI were compared with expressed genes in iPS cell-derived podocytes, kidney biopsy isolated human glomeruli, and mouse podocytes (Figure 4H), as reported by Sharmin et al. [48]. This analysis revealed that many key genes associated with podocyte function and structure were upregulated upon differentiation with adv. RMPI, such as *SYNPO*, *NPHS1*, *NPHS2*, *LAMC1*, *INF2*, *SPARC*, *WT1* and *PODXL*.

### 3.5. Effects of Angiotensin II on the Differentiated UM51-Pre-Podo-hTERT Podocytes

We recently showed that angiotensin II (ANGII) treatment in UdRPC-differentiated podocytes triggered disruption of the cytoskeleton, resulting in the inhibition of the podocyte spreading and subsequent loss of foot processes as observed by a round and condensed morphology [41]. To evaluate the effect of ANGII on the differentiated podocytes derived from the UM51-Pre-Podo-hTERT cell line, we treated the cells with 100 µM ANGII for 24 h and with a combination of 100 µM ANGII and 1 µM of the selective, competitive angiotensin II receptor type 1 antagonist Losartan for 24 h. After 24 h, dynamic changes in morphology and a significant downregulation of α–ACTININ expression was observed in cells treated with 100 µM ANGII by immunofluorescence-based detection (Figure 5A). While all untreated podocytes stained positive for α–ACTININ, only a limited number of cells appeared to express α–ACTININ after 24 h of ANGII treatment. Furthermore, the control podocytes showed the typical “fried egg” podocyte morphology, whereas in ANG II treated podocytes, a more rounded and condensed morphology was observed. To confirm that the disruptive effect was indeed mediated by ANGII, the effects on the cytoskeleton were evaluated by gene-specific mRNA expression of ANGII receptors, *AGTR1* and *AGTR2*, as well as *SNYPO* and *NPHS1* (Figure 5B–E). All genes were found to be significantly upregulated (*p* < 0.05) upon 100 µM ANGII for 24 h except *NPHS1*, which was found to be significantly downregulated (*p* = 0.03). These findings indicated that the effect was mediated via both receptors. ANGII treatment upregulated *AGTR1* expression by 0.4-fold (Figure 5B) and *AGTR2* by 0.8-fold (Figure 5C). Interestingly, *SNYPO* was also found to be upregulated by 0.5-fold upon ANGII treatment (Figure 5D), while *NPHS1* was found to be downregulated by 0.4-fold. When cells were treated with a combination of 100 µM ANGII and 1 µM Losartan, the expression of all genes was significantly (*p* < 0.05) upregulated, while the expression of *AGTR2*, *NPHS1* and *SYNPO* was found to be increased by 0.7-, 1.4- and 0.8-fold. Interestingly, mRNA expression of *AGTR1* was significantly upregulated by 24-fold. Furthermore, the Losartan treatment rescued the ANGII-induced disruption of the cytoskeleton as indicated by immunofluorescence-based detection of α–ACTININ expression (Figure 5A). Higher magnification pictures of podocyte cytoskeleton and cytoskeletal changes can be found in Appendix A. Additionally, the significant upregulation of SYNPO (*p* = 0.03) for both treatments was also confirmed by Western blotting (Figure 5E).

## 4. Discussion

Podocytes are a cell type found in the kidney glomerulus, where they have major implications in blood filtration. Since mature podocytes are terminal differentiated cells that are unable to undergo cell division in vivo [13], establishing podocyte cell cultures has been very challenging. In the present manuscript, we have described the successful immortalization of a human podocyte progenitor cell line UM51-PrePodo-hTERT, where the primary cells were isolated directly from the urine of a male of African origin. UM51-PrePodo-hTERT was cultured over the course of one year (~100 passages) with high proliferation capacity. Numerous cell lines have been immortalized by overexpression of hTERT, leading to chromosomal aneuploidies and alterations in gene expression [49,50]. Even though UM51-PrePodo-hTERT has a hypertriploid karyotype, the cells showed contact inhibition and P53 expression and activation, indicating that the cells had not lost cell-cycle checkpoints or undergone cancerous transformation—even though they share a characteristic of malignant cells, which for the most part bear short telomeres, and telomerase activity is observed in 85–90% of all human cancers [34,51].

Comparison of the transcriptomes pertaining to podocytes differentiated from UM51-PrePodo-hTERT with previously reported datasets of iPSC–derived podocytes, biopsy-derived human glomeruli, and mouse podocytes [48], revealed distinct clustering of UdRPCs and their differentiated counterparts, whereas most of the genes were found to be upregulated after our podocyte differentiation protocol. These results confirmed the differentiation capacity of UM51-PrePodo-hTERT into mature podocytes—based on both protein and mRNA expression of LMX1B, NPHS1, SYNPO and WT1. Interestingly, by employing our recently published podocyte differentiation protocol [41], the proliferation capacity of UM51-PrePodo-hTERT was diminished and a significant upregulation of SYNPO expression was observed. SYNPO has been recognized as a key marker of differentiated post-mitotic podocytes [15,17,18], which cannot be detected in undifferentiated or dedifferentiated cells [19,20]. Furthermore, cells in the glomerulus expressing CD133, CD24 and CD166 have been recognized as being highly proliferative, while cells expressing only CD133 and CD24 show lower proliferative capacity and a committed phenotype towards differentiation [52]. Noteworthily, we detected a downregulation of CD24 after immortalization. A study carried out by Paranjape et al. introducing either hTERT or SV40 into human cells, also reported a generation of a cell population with lower levels of CD24. Therefore, we concluded that the observed CD24 downregulation was caused by the introduction of hTERT and subsequently the immortalization process [46]. Taken together, these two findings, expression of SYNPO and loss of CD24 by cultivation of the UM51-PrePodo-hTert line in adv. RPMI and RA, confirm the differentiation potential of this cell line into podocytes.

We propose that UM51-PrePodo-hTERT can: (a) be differentiated into mature podocytes, and (b) exit the hTERT-driven cell-cycle progression and enter the G_O_ phase. Of note, during nephron development, podocytes lose their mitotic activity [4] and SYNPO expression is first detectable during the capillary loop stage [15]. This transition is reflected by our KI-67 and SYNPO stainings in the UM51-hTERT-Pre-Podo cells cultured in adv. RPMI.

Finally, we evaluated the responsiveness of UM51-PrePodo-hTERT to angiotensin II (ANGII) which is a key mediator of the renin–angiotensin system (RAS) [1,41]. Elevated levels of ANGII have been identified as a main risk factor for the initiation and progression of chronic kidney disease (CKD). Increased ANGII concentrations are associated with the downregulation of nephrin and synaptopodin expression in podocytes [53,54], leading to podocyte injury [55] and subsequent apoptosis [56]. We observed a disruptive effect of ANGII on the cytoskeleton of podocytes derived from UM51-PrePodo-hTERT, which was accompanied by upregulation of the two angiotensin II receptors, AGTR1 and AGTR2. Here, it is interesting to note that most cells completely downregulated α-ACTININ expression while SYNPO expression was found to be upregulated. This might be explained by the duration of the differentiation. The cells were cultured in adv. RPMI for seven days, in which not all cells entered the quiescent state, as indicated by our KI-67 staining and proliferation assay. This makes it tempting to speculate that the hTERT-driven immortalization sustains a subpopulation of the immortalized cells in a proliferative state, which, then triggered by the culture condition, enables the podocyte lineage a specific gene expression repertoire.

There are currently numerous angiotensin receptor blockers (ARBs) and angiotensin-converting enzyme inhibitors (ACEIs) in clinical use for the treatment of a variety of renal diseases [57,58]. A universal ARB is Losartan, an anti-hypertensive agent. Losartan is used for the management of hypertension, as well as for other therapeutic indications. Its action is based on reversing the effects of ANG II on AGTR1 by the selective masking of AGTR1. Our data provide evidence that the addition of Losartan can counteract the downregulation of the slit diaphragm protein NPHS1 induced by ANG II, thus, leading to the restoration of podocyte architecture by the upregulated expression of nephrin. From this, it can be deduced that Losartan can efficiently reduce ANGII binding to AGTR1.The activation of AGTR2 has been linked to vasodilation, development, cell differentiation, tissue repair and apoptosis [59,60]. Therefore, the masking of AGTR1 and, therefore, the hypothesized hyperactivation of AGTR2, might restore the expression levels of podocyte-specific genes back to the differentiated healthy state.

To summarize, we have described the successful immortalization of the very first male African Pre-Podocyte cell line—UM51-Pre-Podo-hTERT—which grew as a monolayer, and showed contact inhibition and a cobblestone-like morphology typical for epithelial cells. UM51-Pre-Podo-hTERT can be: (a) efficiently transfected as shown by GFP-encoding vector expression, and (b) differentiated into mature podocytes, thus, is amenable for studying nephrogenesis and associated diseases, thus, obviating the need of iPSCs. Furthermore, the responsiveness of the derived podocytes to ANGII implies potential applications in kidney-associated disease modelling, nephrotoxicity studies and drug screening.

## Figures and Tables

**Figure 1 cells-12-00342-f001:**
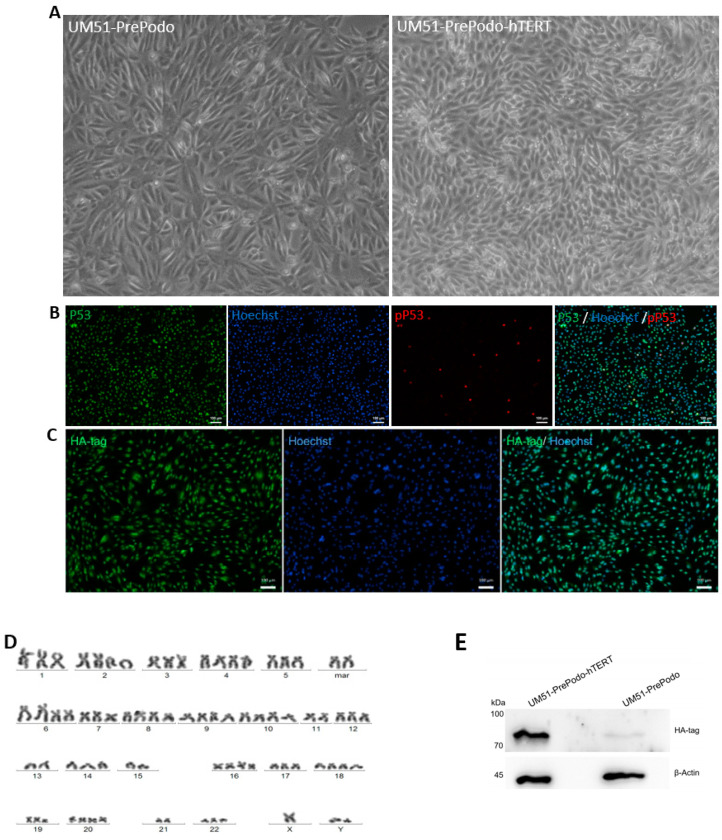
Successful immortalization of the UM51-PrePodo cell line. The UM51-PrePodo cell line was successfully immortalized by transfection with the pCDNA-3xHA-hTERT plasmid. Morphology after transfection as observed by light microscopy (**A**). P53 and phosphorylated P53 expression in the immortalized cells were monitored by immunofluorescent-based detection (**B**) (scale bars: 100 µm). Integration of the plasmid was visualized by immunofluorescence-based and Western blot detection of the HA-tag (**C**,**E**). Karyotyping of UM51-PrePodo-hTERT revealed a hypertriploid karyotype with several alterations (**D**). Expression of the hTERT gene was determined by quantitative real-time PCR (**F**). Telomere length was measured by Southern blotting (**G**).

**Figure 2 cells-12-00342-f002:**
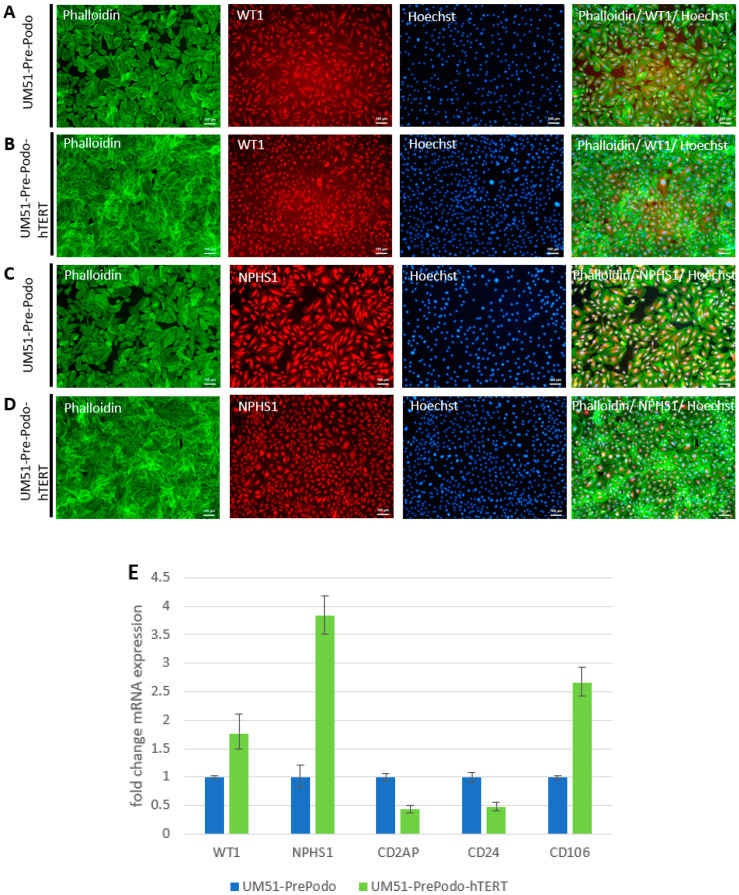
Characterization of immortalized UM51-PrePodo-hTERT. WT1 and NPHS1 expression in the parental UM51-PrePodo as well as the immortalized UM51-PrePodo-hTERT cells were evaluated by immunofluorescence-based detection (**A**–**D**) (scale bars: 100 µm). Immunofluorescence-based detection of the parental UM51-PrePodo is given in (**A**,**C**), while UM51-PrePodo-hTERT is given in (**B**,**D**). Cytoskeleton was stained with phalloidin. mRNA expression of *WT1*, *NPHS1*, *CD2AP*, *CD24* and *CD106* was determined by quantitative real-time PCR (**E**).

**Figure 3 cells-12-00342-f003:**
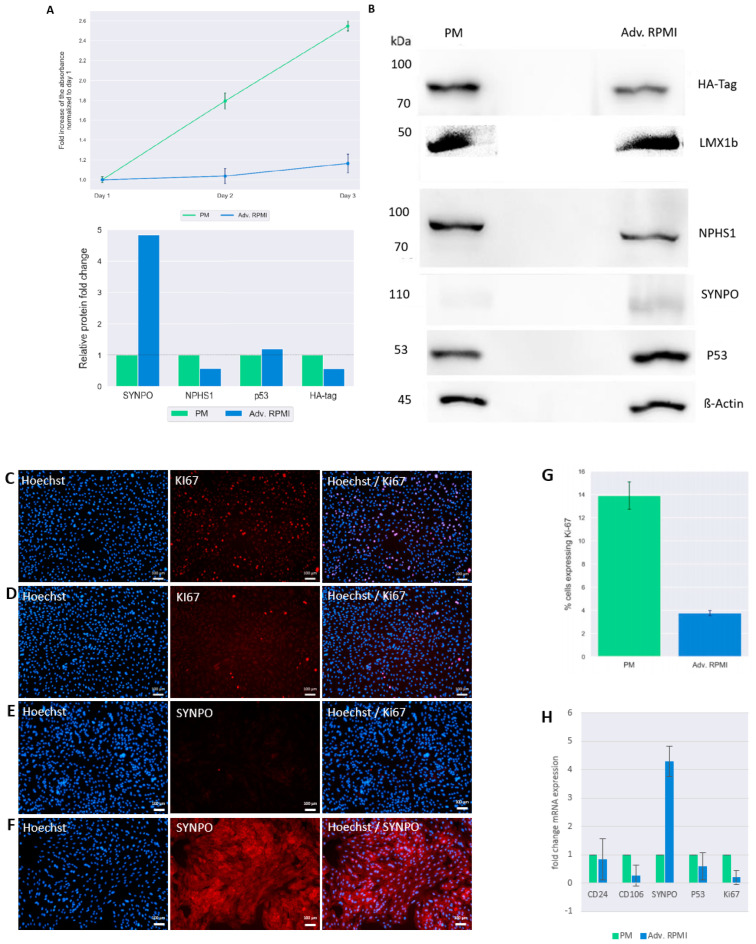
Culturing UM51-PrePodo-hTERT in adv. RPMI initiates podocyte differentiation. Proliferation of immortalized UM51-PrePodo-hTERT cells cultured in PM and adv. RPMI medium was assessed by a resazurin assay (**A**). Relative protein expression normalized to ß-ACTIN for the HA-tag, LMX1B, NPHS1, SYNPO and P53 was detected by Western blot (**B**). Ki-67 and SYNPO expression in UM51-PrePodo-hTERT cultured in PM and adv. RPMI were detected by immunofluorescence-based staining (**C**–**F**). Immunofluorescence-based detection of UM51-PrePodo-hTERT cells grown in PM are given in (**C**,**E**), while cells grown in adv. RPMI are given in (**D**,**F**) (scale bars: 100 µm). Percentage of UM51-PrePodo-hTERT cells cultured in proliferation medium and advanced RPMI expressing the proliferation marker Ki-67 is given in (**G**). mRNA expression of *CD24, CD106, SYNPO, P53 and KI67* was determined by quantitative real-time PCR (**H**).

**Figure 4 cells-12-00342-f004:**
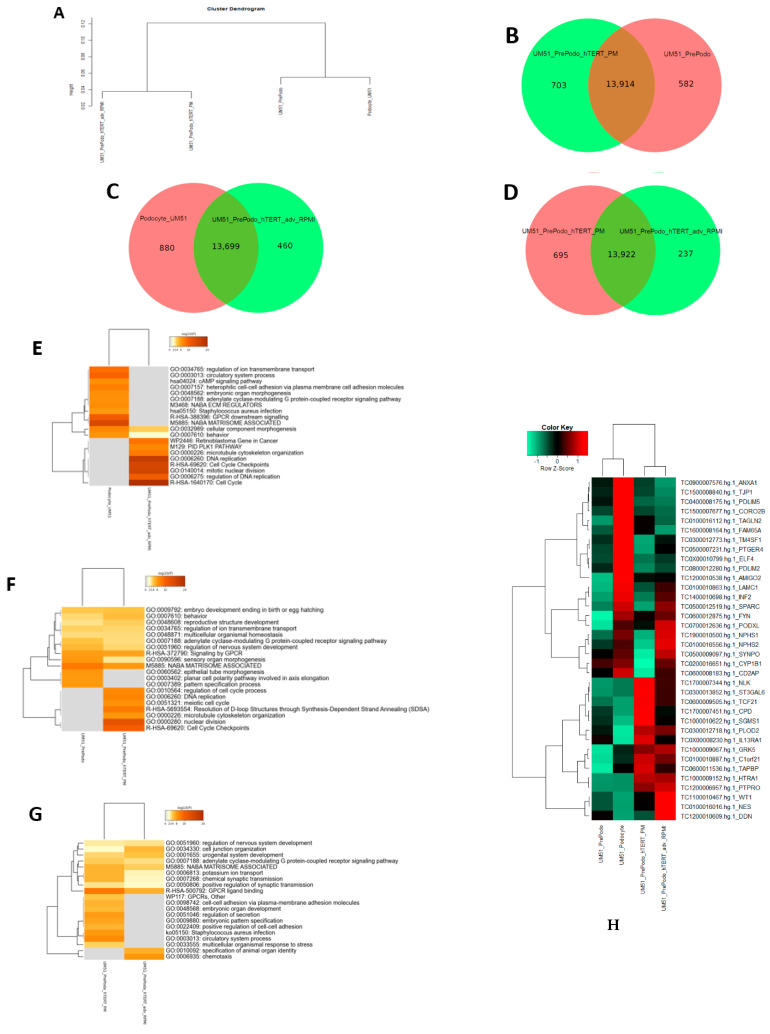
Comparative transcriptome and gene ontology analysis of urine-derived renal progenitor UM51 and derived immortalized UM51-PrePodo-hTERT. A hierarchical cluster dendrogram revealed distinct clusters of immortalized and wild type cells (**A**). Expressed genes (det-p  <  0.05) in UdRPCs and podocytes compared in the Venn diagrams (**B**–**D**), showed distinct (582 in UM51-PrePodo; 703 in UM51-PrePodo-hTERT in PM; 880 in Podocyte UM51; 460 in UM51-PrePodo-hTERT in adv. RPMI; 605 in UM51-PrePodo-hTERT in PM; and 237 UM51-PrePodo-hTERT in adv. RPMI) and overlapping (13,914, 13,699 and 13,922) gene expression patterns. The most over-represented GO BP-terms exclusive in either UM51-PrePodo or UM51-PrePodo-hTERT in PM are shown in (**E**). The most over-represented GO BP-terms exclusive in either podocytes UM51 or UM51-PrePodo-hTERT are shown in (**F**) (full gene list can be found in Appendix A). The most over-represented GO BP-terms exclusive in either UM51-PrePodo or UM51-PrePodo-hTERT in adv. RPMI are shown in (**G**) (full gene list can be found in Appendix A). A heatmap comparing UM51-PrePod and UM51-PrePodo-hTERT in PM with their differentiated counterpart UM51-PrePodo-hTERT in adv. RPMI for a gene set commonly expressed in iPS cell-derived podocytes, kidney biopsy isolated human glomeruli and mouse podocytes is shown in (**H**).

**Figure 5 cells-12-00342-f005:**
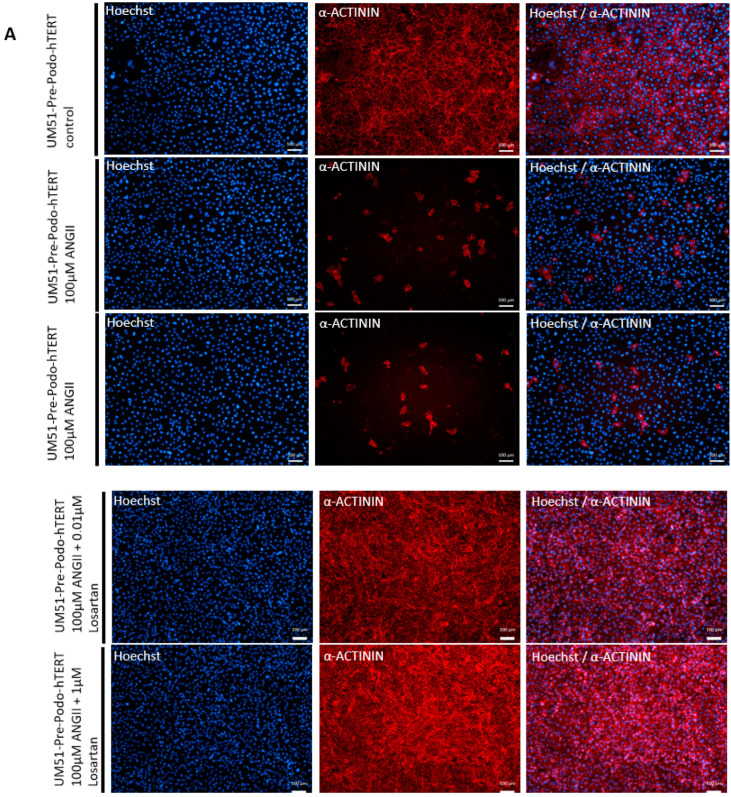
Effects of angiotensin II on the differentiated UM51-Pre-Podo-hTERT cultured in adv. RPMI. UM51-Pre-Podo-hTERT cultured in adv. RPMI supplemented with 30 μM retinoic acid was treated with 100 µM ANGII for 24 h or a combination of 48 h 1 µM Losartan and 24 h 100 µM ANGII. The top panel shows the control morphology. The next two panels show morphology changes after 24 h of 100 µM ANGII treatment, and the last two panels show the morphology for the combination of Losartan and ANGII. The cytoskeleton was visualized by immunofluorescence-based detection of α –ACTININ in red (**A**) (scale bars: 50 µm). Expression of ANGII receptors *AGTR1* (**B**), *AGTR2* (**C**) and expression of podocyte marker *SYNPO* (**D**), and NPHS1 (**E**) were determined by quantitative real-time PCR normalized with the ribosomal encoding gene-RPL0. Expression of the podocyte marker synaptopodin was determined by Western blotting (**F**).

## Data Availability

Raw data are available upon request.

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
