# Peer review of "Derivation of the Immortalized Cell Line UM51-PrePodo-hTERT and Its Responsiveness to Angiotensin II and Activation of the RAAS Pathway"

_cells, 2023, doi:10.3390/cells12030342_

Round 1

Reviewer 1 Report

The Introduction paragraph contains too many informations that are irrelevant to the scope of the study.

Numerous , syntax  ,  grammar errors and typos  make it difficult to understand  some parts of the text.

Methods- RT-PCR: How much RNA/sample has been used?

Lines 62-64: The phenotypical changes occurring between the S-shaped body stage and the capillary loop stage are accompanied by Synaptopodin (SYNPO) [15] and Vimentin [16] expression. Therefore, SYNPO has been recognized as a key marker of differentiated post mitotic podocytes

The sentence is confusing. If both Synpo and Vimentin appear at that stage, why only Synpo but not Vimentin is considered to be a key marker of differentiation?

Supplementary Figure 4:  (B) WB results  for Synpo seem to be questionable. According to the manufacturer’s datasheet, Synpo should run between 100 and 130 kDa , while the image shows a weak band close to 100 kDa and a strong diffused  band between 130 and 180 kDa. This suggests that the antibody preferably reacted with a protein another than synaptopodin. Were the antibodies validated?

Please include a more reliable WB  image. Panel (E) It is hard to guess what the image shows, there is no MW marker. Is it a single or two separate bands?

How many WB analyses  were performed to draw the conclusion that in the Adv RPMI the cells express Synpo?  How many RT-PCR analyses were performed for results shown in fig 3A?

When showing the WB images, please include the information that the image is representative out of (how many?) experiments

Figures 1F, 2, 3, 5: Results are shown as vertical bars +/- SD? SEM? Please add in figure captions the missing informations , including the number of separate experiments

Figure 2: The Pre-Podo hTERT cells seem to be overgrown as compared to Pre-Podo cells . It is difficult to assess the F-actin cytoskeleton in such a mass of cells

Lines 241-243: While real-time PCR based detection revealed a significant increase in  mRNA expression for WT1, NPHS1 and CD106, by 0.77 (p=0,001), 2.83 (p=0,03) and 1.66 -  fold (p=0,05) respectively, a significant downregulation of CD2AP and CD24, by 0.57 (p=0,05) and 0.52 (p=0,05) were observed

It is difficult to understand, in which cells RT-PCR revealed increase or decrease of tested proteins. Immortalized vs non-immortalized?  Figure 2C  clearly shows higher NPHS1 immunofluorescence than in panel D. Does it mean that increased nephrin expression was observed in Pre-podo cells, while NPHS1 mRNA was increased in hTERT cells?

Again, despite of presented  statistical significances (p=0.001 etc) no statistical data is provided, eg number of independent experiments and statistical method.

How the decrease of CD2AP  in immortalized cells can be explained? It rather indicates that the immortalized cells do not fully correspond  to podocytes?

Why have the authors determined LMX1B expression? And what was the reason for testing CD24+ and CD106+? The progenitor cells expressing CD133+, CD24+, CD106+ have a capacity to differentiate into podocytes, but also to tubular epithelial cells. Increased CD106 expression with concomitant drop in CD24 has not been explained or discussed. On the other hand, why typical podocyte markers  such as podocin and podocalyxin have been omitted in the analytical protocol?

Figure 3A : do the bars show the results of a single experiment? Why are the SD (SEM) lacking?

Figures 2, 3 – results of  which passage are shown?

The Authors applied 100 micromolar Ang II which is extremely high concentration. While in human plasma AngII concentrations were found in picomolar ranges, in the in vitro experiments usually nanomolar concentrations have been used. Please explain why such high Ang II concentration has been applied and please relate it to the possible effects due to AngII overdosing.

Please explain why have you used 1 micromolar Losartan when AngII concentration was 100 times higher? Usually, inhibitor/blocker concentration is at least equal if not higher than the agonist concentration. Otherwise you can not be sure that the receptor has been fully blocked.

Figure 5: Changes in cell morphology and cytoskeleton are primarily demonstrated by actin reorganization. Why F-actin staining is not included? Additionally, to prove the effects of AngII on the cell morphology, the  authors should also include the image with higher magnification,  clearly showing details in a single cell.  

The authors claim that the UM51- 428PrePodo-hTERT cell line maintained the podocyte phenotype over 1 year (100 passages= of culture. However, in the manuscript there is no proof for this statement. The Authors have not presented data comparing the properties of the cells from early and late passages.

In contrast to primary podocytes directly obtained from glomeruli, podocytes excreted in urine can be impaired by the factor that caused their detachment from GBM.  The authors have performed some basic analyses to confirm the origin and phenotype of UM51-PrePodo-hTERT but how can they be sure that it is not an immortal defective cell line?

Author Response

Numerous, syntax,  grammar errors and typos  make it difficult to understand  some parts of the text.

We have edited the English language.

Methods- RT-PCR: How much RNA/sample has been used?

Material and methods have been rephrased and now reads:

1µg of RNA was used as Input for reverse transcription in a volume of 10µl. Real time measurements were carried out on the Step One Plus Real Time PCR Systems using 1:10 diluted template cDNA in a MicroAmp Fast optical 384 Well Reaction Plate and Power Sybr Green PCR Master Mix (Applied Biosystems, Foster City, United States).

Lines 62-64: The phenotypical changes occurring between the S-shaped body stage and the capillary loop stage are accompanied by Synaptopodin (SYNPO) [15] and Vimentin [16] expression. Therefore, SYNPO has been recognized as a key marker of differentiated post mitotic podocytes

The sentence is confusing. If both Synpo and Vimentin appear at that stage, why only Synpo but not Vimentin is considered to be a key marker of differentiation?

This has been rephrased as; Therefore, both proteins are recognized as key markers of differentiated post-mitotic podocytes.

Supplementary Figure 4:  (B) WB results  for Synpo seem to be questionable. According to the manufacturer’s datasheet, Synpo should run between 100 and 130 kDa , while the image shows a weak band close to 100 kDa and a strong diffused  band between 130 and 180 kDa. This suggests that the antibody preferably reacted with a protein another than synaptopodin. Were the antibodies validated?

Antibodies were purchased from Thermo Fisher https://www.thermofisher.com/antibody/product/SYNPO-Antibody-Polyclonal/PA5-56997

The company validated the antibody using human skeletal muscle tissue. The image presented on their webpage also shows multiple bands, even though they are all smaller than 100kda. We cannot exclude that the antibody binds with another protein, but nevertheless we are convinced that Synaptopodin is expressed after our differentiation protocol. - See the heatmap (fig. 4h) of podocyte-associated genes and the PCR data in Fig. 2e.

Please include a more reliable WB image. Panel (E) It is hard to guess what the image shows, there is no MW marker. Is it a single or two separate bands?

The image has been removed, MW marker has been added and there are indeed two bands.

How many WB analyses were performed to draw the conclusion that in the Adv RPMI the cells express Synpo?  How many RT-PCR analyses were performed for results shown in fig 3A?

When showing the WB images, please include the information that the image is representative out of (how many?) experiments

We have included a paragraph in the material and methods section about statistics:

Data are presented as arithmetic means + standard error. In total, three independent experiments were performed and used for the calculation of mean values. Statistical significance was calculated using the two-sample Student’s t-test with a significance threshold p = 0.05.

Figures 1F, 2, 3, 5: Results are shown as vertical bars +/- SD? SEM? Please add in figure captions the missing informations , including the number of separate experiments

We have included a paragraph in the material and methods section about statistics:

Data are presented as arithmetic means + standard error. In total, three independent experiments were performed and used for the calculation of mean values. Statistical significance was calculated using the two-sample Student’s t-test with a significance threshold p = 0.05.

Figure 2: The Pre-Podo hTERT cells seem to be overgrown as compared to Pre-Podo cells . It is difficult to assess the F-actin cytoskeleton in such a mass of cells.

hTERT drives proliferation, hence the observed overgrowth compared to the pre-podo cells

Lines 241-243: While real-time PCR based detection revealed a significant increase in  mRNA expression for WT1, NPHS1 and CD106, by 0.77 (p=0,001), 2.83 (p=0,03) and 1.66 -  fold (p=0,05) respectively, a significant downregulation of CD2AP and CD24, by 0.57 (p=0,05) and 0.52 (p=0,05) were observed

It is difficult to understand, in which cells RT-PCR revealed increase or decrease of tested proteins. Immortalized vs non-immortalized? 

This has been specified in the text as follows:

The comparison between the primary and the immortalized cell line by real-time PCR based detection revealed a significant increase in mRNA expression for WT1, NPHS1 and CD106, by 0.77 (p=0,001), 2.83 (p=0,03) and 1.66 - fold (p=0,05) respectively, and a significant downregulation of CD2AP and CD24, by 0.57 (p=0,05) and 0.52 (p=0,05) in the immortalized cells.

 Figure 2C clearly shows higher NPHS1 immunofluorescence than in panel D. Does it mean that increased nephrin expression was observed in Pre-podo cells, while NPHS1 mRNA was increased in hTERT cells?

Yes this is correct.

Again, despite of presented  statistical significances (p=0.001 etc) no statistical data is provided, eg number of independent experiments and statistical method.

We have added a paragraph in the material and methods section about statistics:

Data are presented as arithmetic means + standard error. In total, three independent experiments were performed and used for the calculation of mean values. Statistical significance was calculated using the two-sample Student’s t-test with a significance threshold p = 0.05.

How the decrease of CD2AP in immortalized cells can be explained? It rather indicates that the immortalized cells do not fully correspond to podocytes?

Indeed, the immortalized cells do express less CD2AP compared to the primary cells. We have included CD2AP in the heatmap in figure 4 confirming the PCR data. The heatmap in figure 4 also shows that the expression of key podocyte markers, such as NPHS1, NPHS2, PODXL and SNYPO is even higher in the immortalized cells, therefore we are convinced that the immortalized cells are indeed podocytes.

Why have the authors determined LMX1B expression? And what was the reason for testing CD24+ and CD106+? The progenitor cells expressing CD133+, CD24+, CD106+ have a capacity to differentiate into podocytes, but also to tubular epithelial cells. Increased CD106 expression with concomitant drop in CD24 has not been explained or discussed. On the other hand, why typical podocyte markers such as podocin and podocalyxin have been omitted in the analytical protocol?

We have modified the discussion:

Furthermore, cells in the glomerulus expressing CD133, CD24 and CD166 have been recognized to be highly proliferative, while cells expressing only CD133 and CD24 show lower proliferative capacity and a committed phenotype towards differentiation [52]. Noteworthy, we detected a downregulation of CD24 after immortalization. A study carried out by Paranjape et al., introducing either hTERT or SV40 into human cells, also reported a generation of a cell population with lower levels of CD24. Therefore, we conclude that the observed CD24 downregulation is caused by the over expression of hTERT and subsequently the immortalization process [46].

Figure 3A : do the bars show the results of a single experiment? Why are the SD (SEM) lacking?

Figures 2, 3 – results of  which passage are shown?

We have added a paragraph in the material and methods section about statistics:

Data are presented as arithmetic means + standard error. In total, three independent experiments were performed and used for the calculation of mean values. Statistical significance was calculated using the two-sample Student’s t-test with a significance threshold p = 0.05.

The Authors applied 100 micromolar Ang II which is extremely high concentration. While in human plasma AngII concentrations were found in picomolar ranges, in the in vitro experiments usually nanomolar concentrations have been used. Please explain why such high Ang II concentration has been applied and please relate it to the possible effects due to AngII overdosing.

In our previous published manuscript “Erichsen, L.; Thimm, C.; Bohndorf, M.; Rahman, M.S.; Wruck, W.; Adjaye, J. Activation of the Renin-Angiotensin System Disrupts the Cytoskeletal Architecture of Human Urine-Derived Podocytes. Cells 2022, 11, 1095, doi:10.3390/cells11071095.”, we successfully tested and stimulated the podocytes cells with a 100µM concentration of ANGII and observed a massive disruption of the cytoskeleton. To follow up on these experiments and to show that the immortalized cell line responds in the same way, we decided to apply the same concentration of ANGII to the cells.

Please explain why have you used 1 micromolar Losartan when AngII concentration was 100 times higher? Usually, inhibitor/blocker concentration is at least equal if not higher than the agonist concentration. Otherwise you can not be sure that the receptor has been fully blocked.

We evaluated several concentrations of Losartan with regards to cell morphology and Angiotensin receptor1 expression (manuscript in preparation). From our results the combination of 1µM Losartan and 100µM ANGII performed the best with regards to the aforementioned parameters.

Figure 5: Changes in cell morphology and cytoskeleton are primarily demonstrated by actin reorganization. Why F-actin staining is not included? Additionally, to prove the effects of AngII on the cell morphology, the  authors should also include the image with higher magnification,  clearly showing details in a single cell.  

Supplementary figure 5 shows higher magnification images of the cell morphology after ANGII treatment. Unfortunately, we do not have the facility to provide single cell images.

The authors claim that the UM51- 428PrePodo-hTERT cell line maintained the podocyte phenotype over 1 year (100 passages= of culture. However, in the manuscript there is no proof for this statement. The Authors have not presented data comparing the properties of the cells from early and late passages.

All experiments have been carried out in the past two years and were done at least in duplicates. During that time we did not observe a difference between early and late passages of the immortalized cells. The aim of the manuscript is to described and characterize the cell line as podocyte progenitor cell line. We are convinced that early as well as late passages  are both capable of differentiating into podocytes.   

In contrast to primary podocytes directly obtained from glomeruli, podocytes excreted in urine can be impaired by the factor that caused their detachment from GBM.  The authors have performed some basic analyses to confirm the origin and phenotype of UM51-PrePodo-hTERT but how can they be sure that it is not an immortal defective cell line?

Despite the shown aberrant karyotype, the cells stop proliferating and express typical podocyte markers when cultured according our published differentiation protocol. Furthermore, they show contact inhibition, and they are responsive to ANGII in the same way as the primary cells. For these reasons we are certain that the primary cells have been successfully immortalized and do not represent an immortal defective cell line.

Reviewer 2 Report

Several times the authors specify the source of the initial cells selected for immortalisation. However, no supplemental data are brought concerning the reason for the selection of the donor.

Potential associated pathologies, reasons for the presence of the viable podocyte cells in the urine of that particular patient, if other patients could have been selected as source for the initial cells and so on.

Also, if podocytes are a common occurrence in the urine and in what amounts.

How was the identification made of the progenitor cells in the urine samples?

Otherwise, the paper is excellent, the amount of work and dedication involved is impressive.

Round 2

Reviewer 1 Report

One of my former was  The authors claim that the UM51- 428PrePodo-hTERT cell line maintained the podocyte phenotype over 1 year (100 passages= of culture. However, in the manuscript there is no proof for this statement. The Authors have not presented data comparing the properties of the cells from early and late passages.

 The Authors ‘ response is that they are convinced that over 100 passages, there were no changes in podocyte phenotype. However, in podocyte culture this is not a commonly observed phenomenon . Therefore, I strongly recommend that the Authors include some images (e.g. IF, WB) for early and late (e.g. 10 vs 100) passages  in supplementary material. This could be a valuable support for other researchers.

Apart from this single remark, I have no further concerns .

Author Response

One of my former was  The authors claim that the UM51- 428PrePodo-hTERT cell line maintained the podocyte phenotype over 1 year (100 passages= of culture. However, in the manuscript there is no proof for this statement. The Authors have not presented data comparing the properties of the cells from early and late passages.

 The Authors ‘ response is that they are convinced that over 100 passages, there were no changes in podocyte phenotype. However, in podocyte culture this is not a commonly observed phenomenon . Therefore, I strongly recommend that the Authors include some images (e.g. IF, WB) for early and late (e.g. 10 vs 100) passages  in supplementary material. This could be a valuable support for other researchers.

Apart from this single remark, I have no further concerns .

We have added a comparison of Synpo expression as supplementary figure 6 between UM51-hTERT podocytes passage 3 and passage 75.
